# The pro-inflammatory effect of Staphylokinase contributes to community-associated *Staphylococcus aureus* pneumonia

Yanan Wang [1,3], Na Zhao[1,3], Ying Jian[1], Yao Liu[1], Lin Zhao[1], Lei He [1], Qian Liu [1✉] & Min Li [1,2✉]

Pneumonia caused by community-associated *Staphylococcus aureus* (CA-SA) has high morbidity and mortality, but its pathogenic mechanism remains to be further investigated. Herein, we identify that staphylokinase (SAK) is significantly induced in CA-SA and inhibits biofilm formation in a plasminogen-dependent manner. Importantly, SAK can enhance CA-SA-mediated pneumonia in both wild-type and cathelicidins-related antimicrobial peptide knockout (*CRAMP$^{-/-}$*) mice, suggesting that SAK exacerbates pneumonia in a CRAMP-independent manner. Mechanistically, SAK induces pro-inflammatory effects, especially in the priming step of NLRP3 inflammasome activation. Moreover, we demonstrate that SAK can increase K$^+$ efflux, production of reactive oxygen species production, and activation of NF-κB signaling. Furthermore, the NLRP3 inflammasome inhibitor can counteract the effective of SAK induced CA-SA lung infection in mice. Taken together, we speculate that SAK exacerbates CA-SA-induced pneumonia by promoting NLRP3 inflammasome activation, providing new insights into the pathogenesis of highly virulent CA-SA and emphasizes the importance of controlling inflammation in acute pneumonia.

[1] Department of Laboratory Medicine, Ren Ji Hospital, Shanghai Jiao Tong University School of Medicine, Shanghai, China. [2] Faculty of Medical Laboratory Science, College of Health Science and Technology, Shanghai Jiao Tong University School of Medicine, Shanghai, China. [3] These authors contributed equally: Yanan Wang, Na Zhao. ✉email: qq2005011@163.com; ruth_limin@126.com

Staphylococcus aureus (S. aureus) is one of the most important pathogens and can cause a wide variety of infections[1,2]. During the past decade, S. aureus has become an important cause of community acquired pneumonia[3–5]. In contrast with hospital-associated S. aureus (HA-SA), which usually infects patients with impaired immunity, community-associated S. aureus (CA-SA) infections can occur in other healthy individuals, suggesting that these bacterial strains have greater virulence[6].

High expression of virulence genes that promote invasive infections plays an important role in the pulmonary pathology associated with CA-SA related pneumonia[5,7,8]. However, excessive local inflammation and tissue damage can also impede bacterial clearance[9]. Mouse pneumonia models lacking innate immune signal components showed significantly improved outcomes when infected by S. aureus[10,11]. The NLRP3 inflammasome is essential for the host's immune defense against bacterial, fungal and viral infections[12]. But, when NLRP3 is dysregulated or overactivated, the activated Caspase-1 promotes the maturation of IL-1β and IL-18, facilitates the recruitment of neutrophils and participates in the inflammatory process of acute lung injury damage[9,13–15].

Recent evidence suggests that the epidemic HA-SA clones in China mainly belong to ST5 and ST239, while the main prevalent clones of CA-SA include ST59, ST398, ST188 and ST1[16–18]. The ST398 clone was previously considered to be livestock-associated, but more and more epidemiological studies have shown that ST398 isolates spread in community populations are usually not related to animal infections[19]. The isolation rate of ST398 continues to increase, and it has become one of the most prevalent CA-SA clones in Shanghai, China[16,19,20]. We have reported the effects of high expression levels of virulence genes such as the ESAT-6 secretion system on the pathogenicity of ST398[21,22]. In addition, prophage also plays an important role in bacterial pathogenicity and evolution[23,24]. The gene encoding staphylokinase (SAK) is located on prophage 3[25] and it is conserved in human adapted S. aureus (HO-SA)[19]. SAK was originally found to promote the activation of plasminogen and exert its fibrinolytic function in specific hosts[26,27]. Moreover, studies have shown that SAK can bind to α-defensins[28], human cathelicidin LL-37 and mouse cathelicidins-related antimicrobial peptide (CRAMP)[29] to regulate fibrinolysis and evade innate immunity defenses. While the research on SAK has often focused on its fibrinolytic function, the role of SAK in the pathogenesis of S. aureus remains insufficient and controversial.

In the present study, our results showed increased expression of SAK in ST398 and ST59 isolates, pointing to a potential role of SAK in the virulence of CA-SA. Moreover, we demonstrated a significant role of SAK in ST398 infection-induced acute pneumonia model, which is mainly achieved by activating innate immune signals, especially NLRP3 inflammasome-related pathways.

## Results

### sak is conserved in HO-SA and highly expressed in CA-SA.
Prophage 3 is considered as a potential molecular marker to distinguish livestock-adapted S. aureus (LA-SA) from HO-SA[30]. In the present study, we detected the carrying rate of sak in the predominant HA-SA (ST5, ST239), CA-SA (ST59, ST188, ST398) and LA-SA (ST188, ST398, ST97, ST520) isolates of China. The results showed that 96-100% of HO-SA carried the sak gene, and there was no significant difference between HA-SA and CA-SA, while the detection rate of sak in LA-SA was only 0–28% (Fig. 1a). Among these LA-SA, ST398 and ST188, which carry a higher proportion of sak than ST97 and ST520, can be isolated from both humans and animals. These results further confirmed that the carrier rate of sak is related to human infection.

Although the sak gene is conserved in HO-SA, we found that the expression level of sak in CA-SA was significantly higher than that in HA-SA (Fig. 1b). It has been shown that the increased expression of pathogenicity-related genes is responsible for the high virulence of CA-SA[21,22]. We speculate that the high expression of SAK may also play an important role in promoting the pathogenesis of CA-SA. In addition, we used a chromogenic assay to measure the level of SAK secreted by bacteria in cultures with different incubation times. The results showed that the high SAK secretion level of CA-SA reached its peak in the logarithmic growth phase and remained stable (Fig. 1c). Clinical isolates of S. aureus exhibit similar growth states under in vitro culture conditions (Supplementary Fig. 1a), and we also quantified SAK levels in stationary phase (12 h) supernatants by using standard dilutions of recombinant SAK and found that CA-SA secreted SAK concentrations of approximately 3 μg/ml, at least 5-fold higher than HA-SA (Supplementary Fig. 1b).

### SAK negatively regulates the biofilm formation in plasminogen-dependent manner.
SAK can promote the activation of human plasminogen and exert its fibrinolytic function. However, SAK can only activate plasminogen in some specific hosts, including humans, rabbits, and sheep, but not mice[26]. Studies in mice expressing human plasminogen showed that SAK can inhibit the formation of biofilms[26]. Biofilms are involved in antibiotic resistance of bacteria and host immune defenses[31]. But the highly virulent S. aureus often causes invasive infections[21] through high expression of virulence-related genes, and their biofilm formation ability is usually weaker than HA-SA (Fig. 1d).

By detecting the activation effect of SAK on plasminogen, we confirmed the successful construction of sak gene knockout strain and the sak complementary strain (Supplementary Fig. 1c). We found that sak deletion did not affect the growth of ST398 (Supplementary Fig. 1d) and the formation of biofilm in TSB medium (Fig. 1e). Furthermore, the addition of SAK protein did not inhibit the formation of biofilms, but after adding human plasma, significant differences were observed between the wild-type and the sak knockout strain (Fig. 1e). This indicates that the ability of SAK to inhibit biofilm formation is achieved primarily by promoting fibrinolysis in specific hosts, and these results are consistent with the research of Kwiecinski et al[26]. Although CA-SA with high SAK expression showed weak biofilm formation ability (Fig. 1b–d), we speculated that this is more likely due to the high expression level of the important virulence regulator system, such as accessory gene regulator (Agr) system (Supplementary Fig. 2a). High expression of the agr locus can significantly inhibit the formation of biofilms[32], which is consistent with our results (Supplementary Fig. 2b). The Agr system positively regulates a variety of virulence-related genes, and the expression of sak is also regulated by it, but the knockout of sak gene does not affect the expression of agr (Supplementary Fig. 2c, d). Therefore, we speculated that the inhibitory effect of SAK on biofilm formation mainly depends on fibrinolysis.

### SAK of ST398 enhances the severity of acute lung infection in mice.
It is reported that SAK can directly bind to mouse CRAMP, helping to promote fibrinolysis and evasion of the host's innate immunity[29], so we used both wild-type and CRAMP[−/−] mice for study. The lung tissues of ST398-infected wild-type mice showed obvious congestion and edema (Fig. 2a). HE staining of the lung tissues showed that the alveolar structure of mice was destroyed after infection with the ST398 strain, with infiltration of a large number of inflammatory cells (Fig. 2a). Immunohistochemistry analysis of lung tissues also proved that ST398-infected mice had more macrophage infiltration, as shown by increased staining of

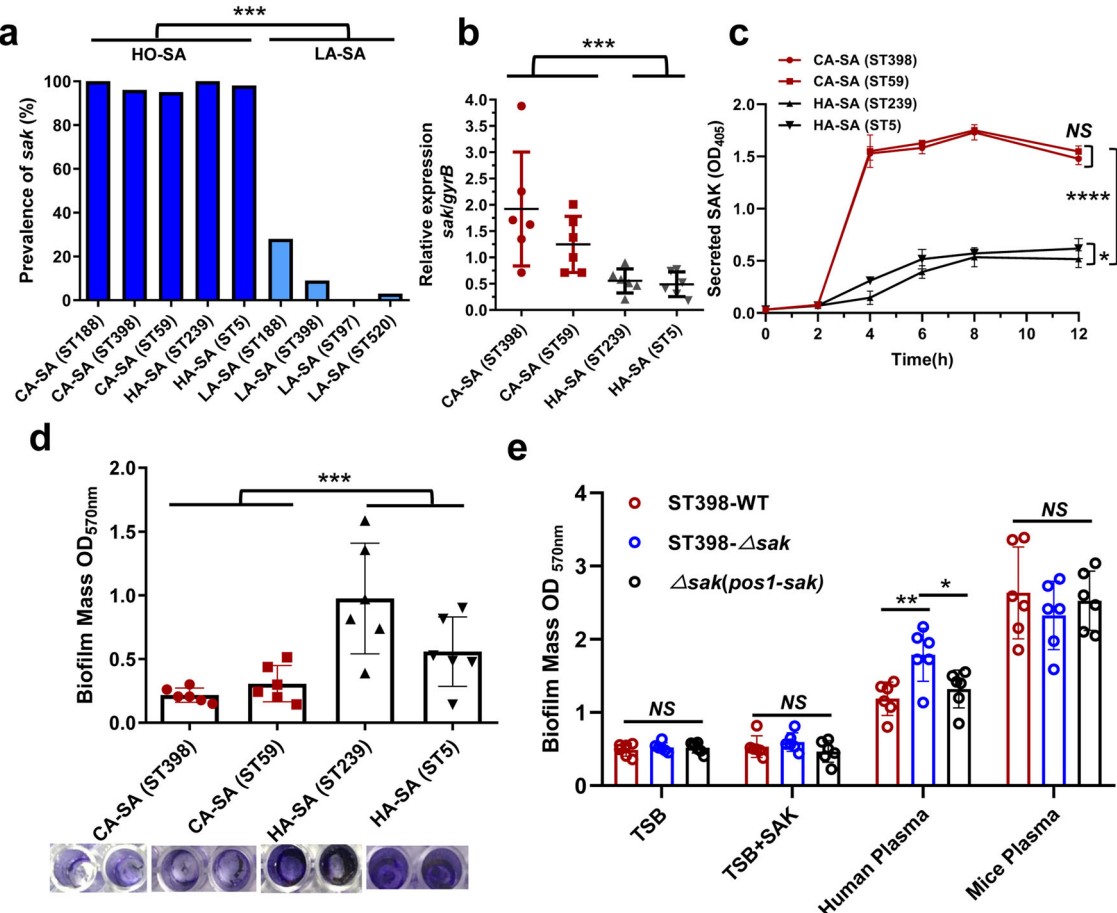

**Fig. 1 SAK is high expressed in CA-SA and can reduce the biofilm formation in plasminogen-dependent manner. a** The prevalence of *sak* gene in *S. aureus* (SA) epidemic isolates. Deep blue: humans-adapted (HO) isolates (including CA-SA and HA-SA); Light blue: livestock-adapted (LA) isolates. Unpaired *t* test was used for statistical analyses between HO-SA and LA-SA after Shapiro–Wilk normality test. **b** qRT-PCR analysis of *sak* gene expression in randomly selected clinical CA-SA (ST398, ST59) and HA-SA (ST239, ST5) isolates, at 4 h of in vitro growth. Relative mRNA levels were calculated using *gyrb* as control and expressed as 2^(−ΔΔCt). Unpaired *t* test was used for statistical analyses between CA-SA and HA-SA after Shapiro–Wilk normality test. **c** Measurement of SAK secretion at different incubation times. Two-way ANOVA with Bonferroni's multiple comparison post-test was used for statistical analyses (at the 12th h). **d** Biofilm formation ability of CA-SA (ST398, ST59) and HA-SA (ST239, ST5). Unpaired *t* test was used for statistical analyses between CA-SA and HA-SA after Shapiro–Wilk normality test. **e** The effect of adding SAK protein, human plasma and mice plasma on the biofilm formation of ST398-WT, *sak* deletion mutant isolate and the complemented isolate. Two-way ANOVA with Bonferroni's multiple comparison post-test was used for statistical analyses. All data in Fig. 1 are presented as mean ± SD and *$p < 0.05$, ** $p < 0.01$, *** $p < 0.001$.

F4/80, a well characterized macrophage marker (Fig. 2b). Furthermore, the wet weight of the lungs and the colony-forming unit (CFU) counts in lung tissues of ST398-infected mice increased significantly (Fig. 2c, d). Interestingly, similar results can be seen in *CRAMP*$^{-/-}$ mice (Fig. 2e–g). After infection with the *sak* knockout strain, both wild-type mice and *CRAMP*$^{-/-}$ mice had milder lung pathological changes. Therefore, we speculated that SAK can increase the severity of acute lung infection in mice, and the effect of combining with CRAMP is not the main mechanism of SAK's pathogenicity.

**SAK promotes NLRP3 inflammasome-related gene transcription and cytokine release in CA-SA pneumonia.** In order to explore the mechanism of SAK in the process of *S. aureus* infection, we analyzed the RNA sequencing data of lung tissues from wild-type and *sak* gene knockout ST398 infected mice (C57BL/6 wild-type). A total of 580 genes were differentially expressed between the two groups (See Supplementary data 1 for the top 50 significantly different genes). In addition, the results of GSEA analysis identified that genes related to cytokine response,

such as interferon-γ (IFN-γ), interferon-α (IFN-α), interleukin-6 (IL-6), interleukin-1 (IL-1), complement-related genes, and NLRP3 inflammasome-related genes were enriched in the lung tissues of wild-type infected mice (Fig. 3a). Furthermore, we detected 23 cytokines in mouse serum and 8 cytokines in mouse bronchoalveolar lavage fluid (BALF), and the results showed that mice infected with wild-type ST398 had higher levels of most cytokines in both serum (Fig. 3b) and BALF (Fig. 3c, d). In the host's confrontation with pathogens, excessive inflammation can cause tissue damage and poor prognosis. The activation of NLRP3 has been shown to promote cytokine production and cell pyrolysis. We further compared the expression of toll-like receptors and other NLRP3 inflammasome activation-related genes in the two groups. Consistent with the higher bacterial load, nearly all of these molecules are highly expressed in the lung tissues of mice infected with wild-type ST398 (Fig. 3e). In addition, we detected the expression of *IL-1β*, *IFN-α*, *Nlrp3* and *Caspase-1* in lung tissues by qRT-PCR, and the results were consistent with RNA sequencing (Fig. 3f–i). Combined with the bacterial load and pathological changes in mouse lung tissue, we speculate that the lung tissue of mice infected with wild-type

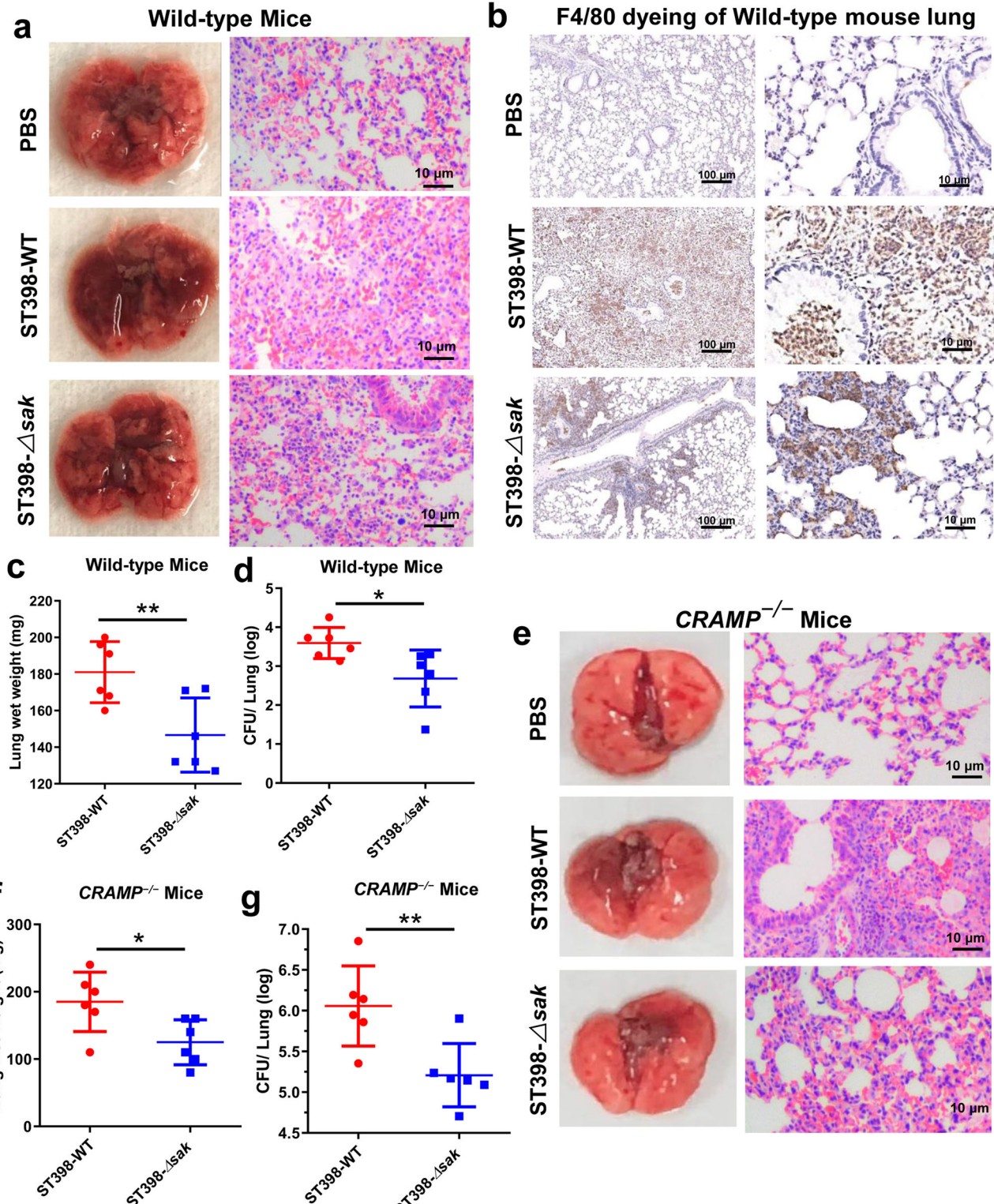

**Fig. 2 SAK enhances CA-SA-mediated lung infection in both wild-type and *CRAMP*[−/−] mice. a–g** $2 \times 10^8$ CFU of *S. aureus* was pipetted into the nares of anesthetized C57BL/6 wild-type mice **a–d** and *CRAMP*[−/−] mice **e–g** (*n* = 6), and mice were euthanized 48 h after inoculation with *S. aureus*.
**a**, **e** Photographs of lungs and corresponding H&E stained sections of mice 48 h after infection. **b** Immunohistochemical staining (F4/80) was performed on the lung tissue of infected mice to observe the infiltration of macrophages. **c**, **f** Lung wet weight of infected mice. **d**, **g** The left lung was homogenized and plated on TSB agar for CFU determination. Unpaired *t* test was used for statistical analyses between ST398-WT and *sak* deletion mutant isolate infected mice after Shapiro–Wilk normality test. Data are presented as mean ± SD and *\*p* < 0.05, *\*\* p* < 0.01.

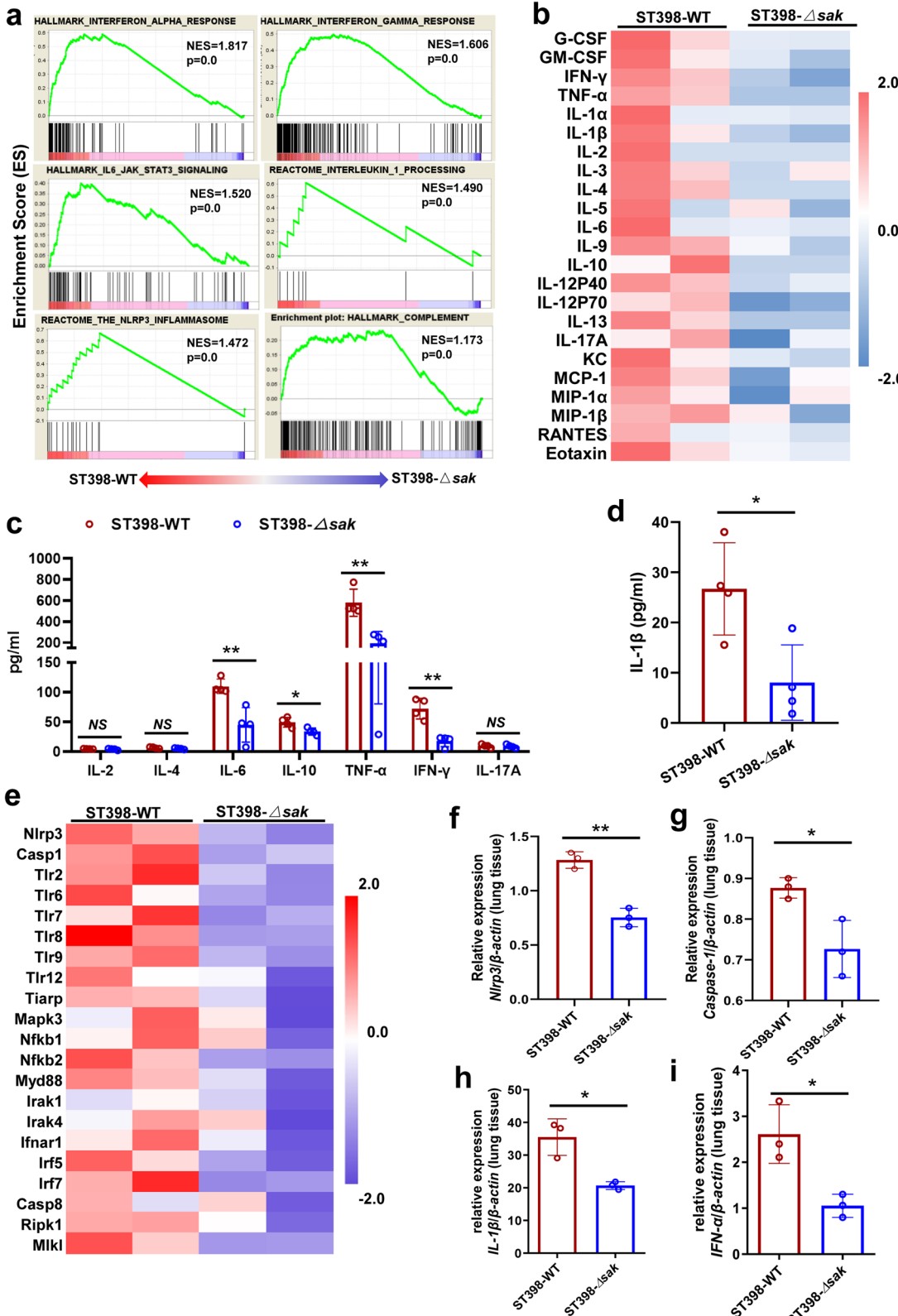

ST398 is in a more active state of inflammatory response than that of *sak* gene knockout ST398 infected mice.

**SAK mainly play a role in the priming step of NLRP3 inflammasome activation and promotes pyroptosis.** We incubated bone marrow-derived macrophages (BMDMs) of mice with

bacterial culture supernatant or SAK protein and extracted the mRNA of the cells. Gene expression was detected by qRT-PCR after reverse transcription, and the results showed that SAK can effectively promote the expression of innate immune inflammation-related factors, such as cytokines (*IL-17A*, *TNF-α*, *IL-1β*) and NLRP3 inflammasome-related genes (Fig. 4a–e). In addition, we assessed cell death by measuring LDH released by

**Fig. 3 SAK promotes NLRP3 inflammasome-related gene transcription and cytokine release in CA-SA pneumonia. a–i** $2 \times 10^8$ CFU of *S. aureus* was pipetted into the nares of anesthetized C57BL/6 wild-type mice. Bronchoalveolar lavage fluid (BALF) was taken 48 h after infection of mice, and total RNA was extracted from lung tissue at the same time. **a** GSEA analysis identified cytokine related genes (*IFN-α, IFN-γ, IL-6, IL-1*), complement related genes and NLRP3 inflammasome pathway enrichment in the ST398-WT infected mice. **b** The level of mouse plasma cytokine was detected by Bio-Plex Pro Mouse Cytokine 23-plex Assay. The data were normalized by the Z score according to the standard deviation from the mean. **c** Cytokine levels in mouse BALF were detected by BD Cytometric Bead Array (CBA) Mouse Th1/Th2/Th17 Cytokine Kit ($n = 4$). Unpaired *t* test was used for statistical analyses after Shapiro–Wilk normality test and data are presented as mean ± SD. **d** Detection of IL-1β in mouse BALF by ELISA ($n = 4$). Unpaired *t* test was used for statistical analyses after Shapiro–Wilk normality test and data are presented as mean ± SD. **e** Comparison of the expression levels of NLRP3 inflammasome activation-related molecules between the two groups. The data were normalized by the Z score according to the standard deviation from the mean. **f–i** qRT-PCR analysis of *Nlrp3, Caspase-1, IL-1β*, and *IFN-α* in the lung tissues of wild-type or *sak* deletion mutants infected mice. Relative mRNA levels were calculated using *β-actin* as control and expressed as 2^(-ΔΔCt). Unpaired *t* test was used for statistical analyses after Shapiro–Wilk normality test and data are presented as mean ± SD. See Supplementary methods for the abbreviated list of cytokines or genes in Fig. 3b, e. *$p < 0.05$, ** $p < 0.01$.

BMDMs after co-incubation with SAK or *S. aureus* supernatants, and the results showed that *sak* gene knockout significantly reduced macrophage death (Fig. 4f).

We further detected the ability of SAK to active NLRP3 inflammasome by western blot. It should be noted that the *sak* gene knockout attenuated the activation of NLRP3 inflammasome by bacterial supernatants, which was reflected by reduced levels of Caspase-1 maturation, gasdermin D (GSDMD) cleavage (Fig. 4g) and decreased IL-1β secretion (Fig. 4h). But we did not observe a significant increase in the activation of NLRP3 inflammasome after SAK protein treatment (Fig. 4g, h, Supplementary Fig. 3a). The classic NLRP3 inflammasome activation is considered to be a two-step process involving priming and activation[33]. We first examined the role of SAK in NLRP3 priming and the results showed increased expression of pro-Caspase-1, GSDMD and pro-IL-1β in SAK-treated BMDMs upon addition of ATP. Furthermore, we could observe the maturation of Caspase-1 and GSDMD, as well as the release of IL-1β, which also indicated that the NLRP3 inflammasome was successfully activated, although the effect was not as strong as that of lipopolysaccharide (LPS) (Fig. 4i, j, Supplementary Fig. 3b). Interestingly, we also found that the addition of SAK could further promote the activation of the NLRP3 inflammasome by LPS (Fig. 4i, j, Supplementary Fig. 3b), implying that SAK may have additive effects with other NLRP3 inflammasome activators, helping to further amplify inflammasome signaling. In addition, we primed the BMDMs for 3 h with LPS and then treated the LPS-primed cells with SAK for determining whether purified SAK can activate NLRP3 inflammasome. Our results suggest a limited role for SAK in the activation step of NLPR3 inflammasome (Supplementary Fig. 3c–e). Although the level of IL-1β release in the SAK-treated group showed an upward trend, the difference was not statistically significant, as was the result of western blot densitometry (Supplementary Fig. 3d, e). Moreover, when the BMDMs of *Caspase-1*$^{-/-}$ mice was incubated with purified SAK or bacterial supernatants, SAK failed to promote GSDMD cleavage and IL-1β maturation (Fig. 4k, l, Supplementary Fig. 3f). Furthermore, by using the NLRP3 inhibitor MCC950, BMDMs of wild-type mice yielded similar results to *Caspase-1*$^{-/-}$ mice (Supplementary Fig. 3g–i). These results suggest that SAK may enhance the host's inflammatory response after *S. aureus* infection and that SAK may play a role in the priming step of NLRP3 inflammasome activation to promote pyroptosis and IL-1β release.

**NLRP3 inflammasome is required for the effective of SAK induced CA-SA lung tissue damage in mice.** We constructed a lethal mouse model of pneumonia to visually show the effect of SAK on the pathogenicity of ST398. The results showed that *sak* gene knockout can improve the survival rate of ST398-infected mice (Fig. 5a). Correspondingly, HE staining of lung tissue

sections of mice infected with wild-type ST398 showed a high degree of inflammatory cell infiltration, and immunohistochemistry analysis showed high levels of IL-1β (Fig. 5b). In addition, the NLRP3 inflammasome can be inhibited by intraperitoneal injection of MCC950 to mice. When mice were given NLRP3 inhibitors, their survival was prolonged, and there was no significant difference in the survival rate of mice infected with wild-type ST398 and *sak* knockout strains (Fig. 5c). This indicates that NLRP3 inflammasome activation plays an important role in SAK's promotion of the pathogenicity of ST398. Furthermore, the inhibitory effect of MCC950 on the NLRP3 inflammasome was confirmed by immunohistochemical staining of IL-1β, and HE staining of lung tissue sections showed that inhibiting NLRP3 inflammasomes can reduce the recruitment of inflammatory cells and weaken the inflammatory damage caused by infection (Fig. 5d).

**SAK promotes NLRP3 inflammasome activation mainly by increasing K$^+$ efflux, reactive oxygen species (ROS) production, and activating NF-κB signaling.** The NLRP3 inflammasome can be activated by a variety of stimuli, such as ion flux, production of ROS, and lysosomal damage[12]. Most NLRP3 inflammasome activators lead to K$^+$ efflux, and in some studies, K$^+$ efflux has even been considered a necessary upstream event for NLRP3 activation[34]. We pretreated cells with the K$^+$ efflux inhibitor glibenclamide and high concentrations of extracellular K$^+$, and then tested whether SAK could effectively promote the expression and release of IL-1β in BMDMs. The results showed that although glibenclamide failed to eliminate the difference in *IL-1β* expression in macrophages caused by the supernatant of ST398 wild-type and *sak* gene knockout strain (Fig. 6a), glibenclamide could inhibit SAK from promoting the secretion of IL-1β from cells (Fig. 6b). Moreover, under the effect of high concentrations of extracellular K$^+$, SAK no longer exhibited additional effects on promoting IL-1β expression and release (Fig. 6a, b). In addition, we also examined intracellular K$^+$ levels and found that the presence of SAK promoted K$^+$ efflux, which could also be inhibited by glibenclamide (Fig. 6c).

Lysosomal disruption results in the release of proteases and phagocytic granules from the lysosome, which promotes the activation of the NLRP3 inflammasome[35]. We pretreated BMDMs with dexamethasone as a stabilizer for lysosomal membranes and found that SAK may not promote IL-1β expression and release by disrupting lysosome (Fig. 6a, b).

NADPH oxidase is one of the main sources of ROS in macrophages, and to explore the role of SAK in ROS production, we used the NADPH oxidase inhibitor apocynin and the ROS scavenger N-acetyl-L-cysteine (NAC) to pretreat BMDMs. The results showed that both apocynin and NAC could effectively inhibit the promoting effect of SAK on the expression and release of IL-1β (Fig. 6a, b). In addition, purified SAK protein could

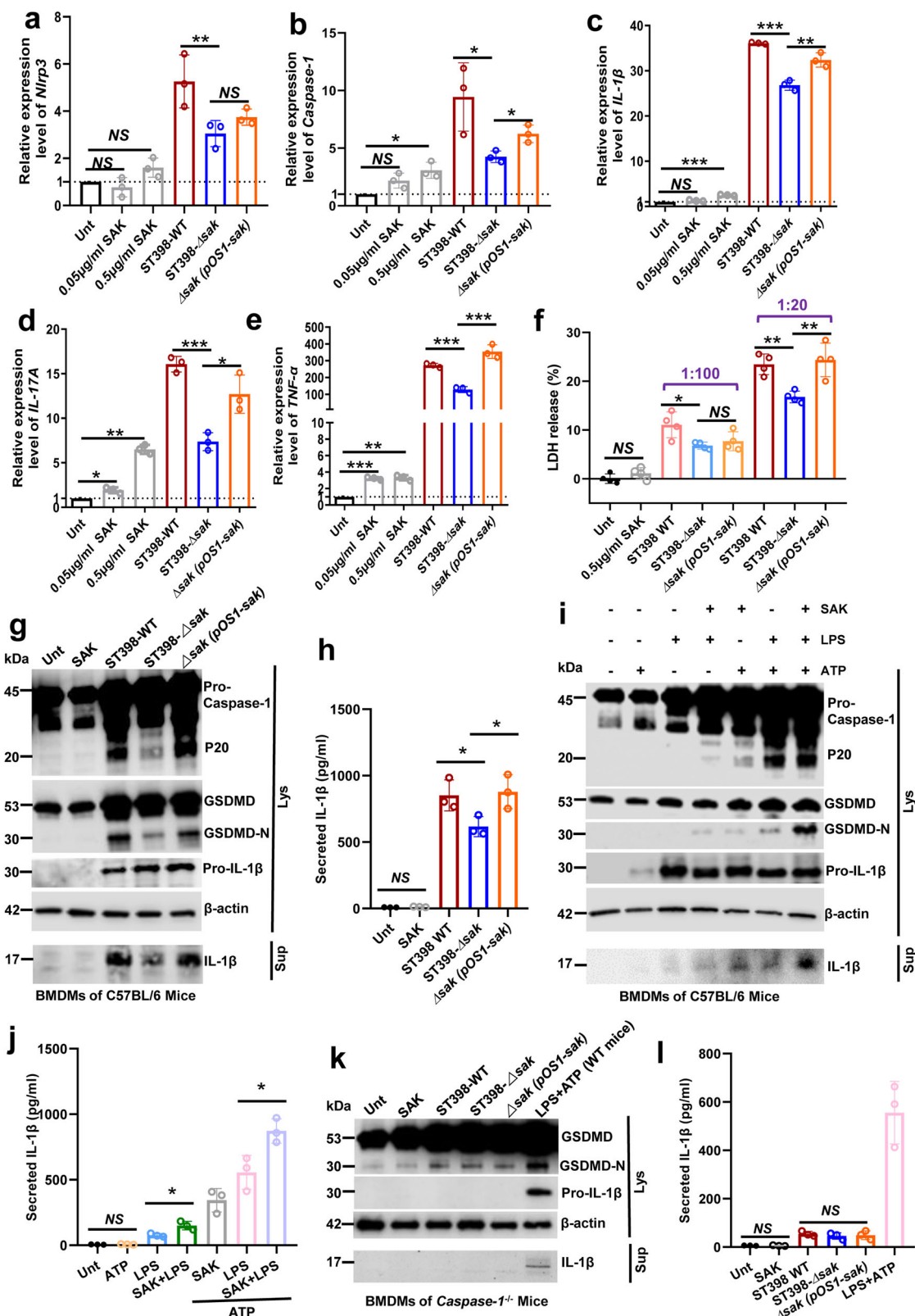

directly promote the production of ROS in BMDMs, and its effect was also inhibited by NAC (Fig. 6d).

Notably, all the above inhibitors significantly inhibited the promoting effect of purified SAK on *IL-1β* expression, we speculate that this result is due to the stronger effect of the inhibitor than SAK and also indicates that purified SAK is not a potent NLRP3 activator, which is consistent with our previous

findings. Furthermore, we measured the cytokines released by human THP-1 cells (human monocyte derived cell line) after stimulation with bacterial culture supernatant. Similar to the results in mouse models, SAK can also stimulate the release of inflammatory factors (IL-1β, TNF-α, IL-17A) in THP-1 cells (Supplementary Fig. 4a–c). Moreover, SAK can also promote the production of ROS in THP-1 cells, and this effect can be inhibited

**Fig. 4 SAK mainly play a role in the priming step of NLRP3 inflammasome activation and promotes pyroptosis. a–e** BMDMs of mice were incubated with bacterial culture supernatant (stationary phase, 1:20 dilution) or SAK (0.05/0.5 μg/ml) for 3 h. The mRNA of the cells was extracted and gene expression relative to the untreated (Unt) group was detected by qRT-PCR after reverse transcription. Relative mRNA levels were calculated using β-actin as internal control and expressed as 2ˆ(-ΔΔCt). Unpaired t-test was used to compare the differences between the variables after Shapiro–Wilk normality test. **f** Determination of macrophage viability by LDH assay. BMDMs were incubated with bacterial culture supernatant (stationary phase, 1:100 or 1:20 dilution) or SAK protein (0.5 μg/ml) for 3 h. Unpaired Student's *t* test was used for statistical analyses of differences between groups after Shapiro–Wilk normality test. **g, h** BMDMs of mice were treated with SAK (0.5 μg/ml) or bacterial supernatants (stationary phase, 1:20 dilution) for 3 h. Immunoblot analysis **g** of cell lysates or culture supernatants. Secreted IL-1β from BMDMs was detected by ELISA **h. i, j** BMDMs of mice were treated with SAK (0.5 μg/ml, 3 h) or LPS (0.2 μg/ml, 3 h), followed by the treatment with ATP (2.5 mM, 30 min). Immunoblots **i** of cell lysates or culture supernatants from mouse BMDMs. Secreted IL-1β from BMDMs was detected by ELISA **j. k, l** BMDMs from *Caspase-1*$^{-/-}$ mice were treated with SAK (0.5 μg/ml, 3 h) or bacterial supernatants (stationary phase, 1:20 dilution, 3 h). Immunoblot analysis **k** of cell lysates or culture supernatants. Secreted IL-1β from BMDMs of *Caspase-1*$^{-/-}$ mice was detected by ELISA **l**. Cell lysates or culture supernatants of wild-type mouse BMDMs treated with LPS (0.2 μg/ml, 3 h) and ATP (2.5 mM, 30 min) were used as positive control. Unpaired *t* test was used to compare the differences between the variables after Shapiro–Wilk normality test. All data are presented as mean ± SD and *$p < 0.05$, ** $p < 0.01$, *** $p < 0.001$.

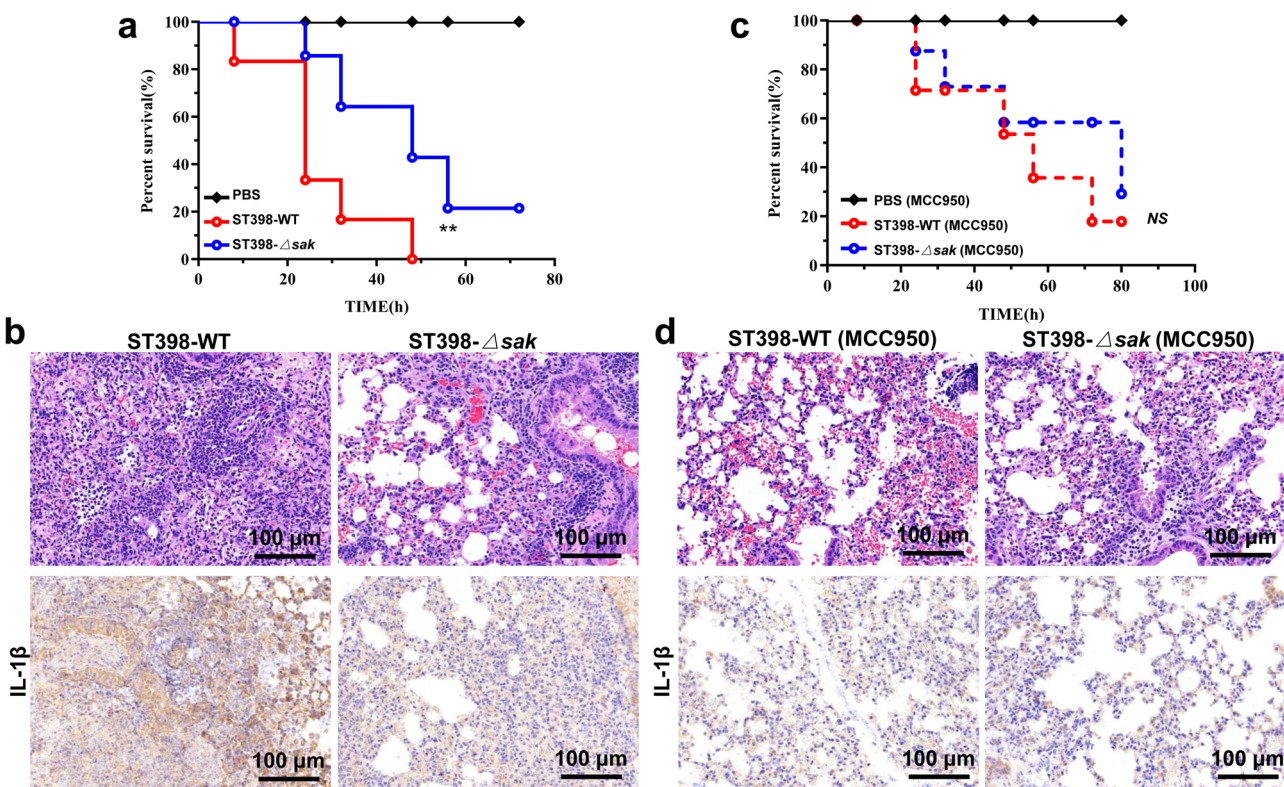

**Fig. 5 Inhibition of the NLRP3 inflammasome can reduce lung infection in mice. a–d** 10$^9$ CFU of *S. aureus* was pipetted into the nares of anesthetized C57BL/6 mice (*n* = 6), and 50 mg/kg MCC950 sodium (NLRP3 inflammasome inhibitor) or PBS was injected intraperitoneally to mice 2 h before infection. **a**, **c** Survival curves were compared using a log-rank (Mantel-Cox) test. **b**, **d** Tissue sections (48 h after infection) for HE staining and immunohistochemical staining (IL-1β). **$p < 0.01$.

by NAC (Supplementary Fig. 4d). This indicates that the activation of the host inflammatory response by SAK is also applicable to human *S. aureus* infections.

Activation of NF-κB can promote the expression of various inflammatory factors and has been shown to play an important role in the activation of NLRP3, especially in the priming step[36]. In the present study, we found that purified SAK could promote the phosphorylation of NF-κB by western blot, and *sak* gene knockout significantly reduced NF-κB activation in ST398 supernatant-stimulated BMDMs (Supplementary Fig. 5a). In addition, the effect of SAK was attenuated after the application of the NF-κB pathway inhibitor BAY 11-7082 (Supplementary Fig. 5b), and there was no statistically significant difference in the release of IL-1β from BMDMs after treatment with the supernatant of ST398 wild strain and *sak* knockout strain (Fig. 6e). These results suggest that SAK may promote the transcription of

inflammation-related factors by activating NF-κB, which may also be one of the mechanisms by which SAK acts on the priming step of the NLRP3 inflammasome.

## Discussion

SAK has been demonstrated to play an important role in the activation of plasminogen in specific hosts[26]. Previous studies showed that SAK can directly bind to human α-defensins[28] and murine CRAMP, and play a role in evading the host's innate immunity[29]. Our results showed that SAK is highly expressed in the highly virulent CA-SA, however, the role of SAK in the pathogenesis of *S. aureus* remains largely unknown. In this study, our data suggest that SAK negatively regulates biofilm formation in a plasminogen-dependent manner. Furthermore, we found that SAK could promote CA-SA-mediated lung infection in the

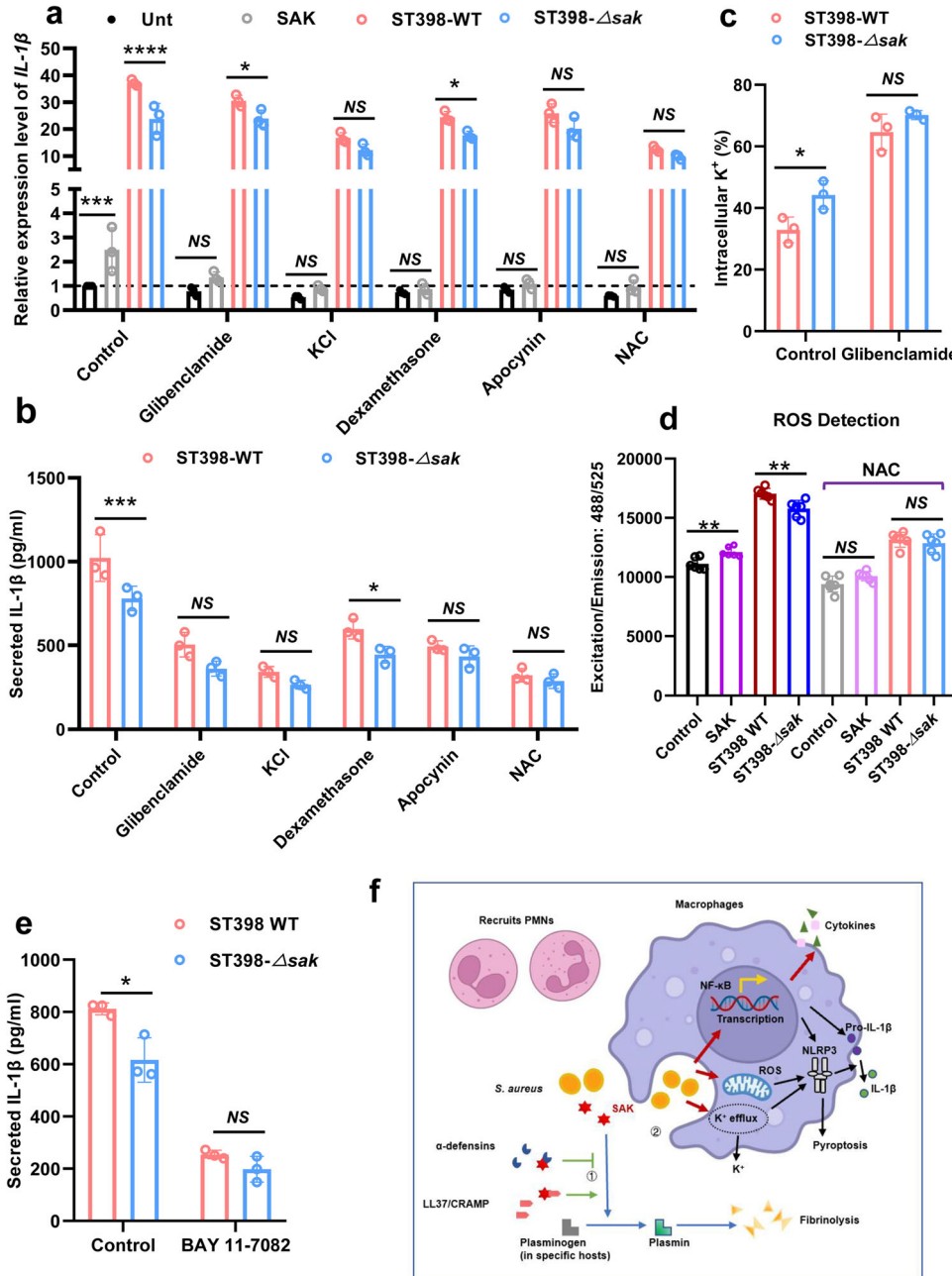

CRAMP-independent manner. In our present study, our results showed that SAK appears to play an important role in acute necrotizing pneumonia by exerting pro-inflammatory effects, particularly by promoting the activation of NLRP3 inflammasome. Mechanistical investigation reveal that SAK contributes the priming step of NLRP3 inflammasome activation through increasing $K^+$ efflux and ROS production, and activating NF-κB signaling (Fig. 6f). Taken together, our findings suggest that SAK exacerbates CA-SA-induced lung infection by promoting NLRP3 inflammasome activation.

The effect of SAK on biofilm formation is human plasminogen-dependent, which is similar with other reports[26,27]. It is worth noting that although the SAK highly expressed CA-SA biofilm formation ability is weak, there is no evidence that SAK can directly inhibit the formation of *S. aureus* biofilm. It has been found that the increased expression of virulence factors controlled by Agr is responsible for the high pathogenicity of ST398, and we speculate that the weak biofilm formation ability of CA-SA is

more likely due to the high expression of the Agr system. In addition, the expression of SAK is also positively regulated by the Agr system. Furthermore, it has been proven that SAK can directly bind to human α-defensins[28] and murine CRAMP, and play a role in evading the host's innate immunity[29]. In this study, our data indicate that high expression of SAK may promote lung infection of ST398 isolates in both wild-type and *CRAMP*[−/−] mice. Due to the limited effect of SAK on plasminogen in mice, we speculate that promoting fibrinolysis and evading host innate immunity may not be the main mechanism of SAK aggravating pneumonia in ST398-infected mice.

Antimicrobial peptides act primarily in the early stages of infection and are thought to be essential innate immune effectors against bacterial infections[37]. In addition, inflammasomes also play important roles in the detection and clearance of pathogens during infection[38]. *S. aureus* has a variety of factors that promote infection, and different infection sites trigger different immune responses in the host. Research shows that NLRP3 signaling plays

**Fig. 6 SAK promotes the activation of NLRP3 inflammasome mainly through increasing K$^+$ efflux and ROS production, and activating NF-κB signaling.**
**a–d** BMDMs were pretreated with glibenclamide (10 μM), high concentrations of extracellular K$^+$ (25 mM), apocynin (200 μM), dexamethasone (200 nM), and NAC (1 mM) for 30 min, and then incubated with SAK (0.5 μg/ml) or bacterial secretion supernatant for 3 h (stationary phase, 1:20 dilution). Two-way ANOVA with Bonferroni's multiple comparison post-test was used to compare the differences between the variables. **a** Relative mRNA levels of *IL-1β* were calculated using *β-actin* as internal control and expressed as 2^(-ΔΔCt). The Y-axis represents the fold change of the *IL-1β* gene expression level in the experimental group relative to the untreated group. **b** The secretion of IL-1β in BMDMs treated with bacterial culture filtrate was detected by ELISA. Two-way ANOVA with Bonferroni's multiple comparison post-test was used to compare the differences between the variables. **c** Detection of intracellular K$^+$ of BMDMs. Two-way ANOVA with Bonferroni's multiple comparison post-test was used to compare the differences between the variables. **d** Detection of ROS production by BMDMs. BMDMs were pretreated with DMSO or NAC (1 mM) for 30 min. Two-way ANOVA with Bonferroni's multiple comparison post-test was used to compare the differences between the variables. **e** Mouse BMDMs were pretreated with BAY 11-7082 (10 μM, 30 min) or DMSO, then cells were treated with SAK (0.5 μg/ml) or bacterial secretory supernatant for 3 h (stationary phase, 1:20 dilution). The secretion of IL-1β in BMDMs was detected by ELISA. Two-way ANOVA with Bonferroni's multiple comparison post-test was used to compare the differences between the variables. All data in Fig. 6 are presented as mean ± SD and *$p < 0.05$, ** $p < 0.01$, *** $p < 0.001$. **f** The role of SAK in *Staphylococcus aureus* infection. ① SAK can neutralize AMPs and regulate fibrinolysis. ②SAK promotes transcription of inflammation-related genes and activation of NLRP3 inflammasome (this study). Research has shown that SAK can promote the activation of plasminogen in specific hosts (human, rabbits, and sheep) and exert its fibrinolytic function (blue arrows). The epithelium expresses antimicrobial peptides to kill extracellular bacteria. Studies have shown that SAK can bind to α-defensins, human cathelicidin LL-37 and mouse CRAMP to regulate fibrinolysis and evade innate immunity defenses (green arrows). In this study, we speculate through preliminary studies that SAK can promote the transcription of inflammatory response-related molecules and the release of cytokines, help recruit polymorphonuclear leukocytes (PMNs) and contribute to a more active state of inflammatory response in the host. Our results suggest that SAK may contribute to the activation of NF-κB and promote K$^+$ efflux and ROS production, thereby playing a role in the activation of the NLRP3 inflammasome (red arrows) and promoting acute lung infection. This figure was drawn by using BioRender (Agreement number: JV23RYUB4V. https://app.biorender.com/) and Photoshop.

an important role in the body's defense against skin and soft tissue infections[38–40]. However, in acute pneumonia caused by highly virulent CA-SA, inflammasome signaling is not necessary for clearance of *S. aureus* from mouse lung tissue[38], and even leads to inflammation-related cell death[41]. In acute lung infections caused by high virulence CA-SA, uncontrolled excessive and long-term inflammation induced by virulence factors can lead to cell death and tissue damage[9,41,42], thereby hindering bacterial clearance[11,43]. In particular, alveolar macrophage loss and immunomodulatory dysfunction caused by Caspase-1-dependent pyroptosis amplify the pathological consequences of infection[13,41]. It is worth noting that RNA sequencing analysis of mouse lung tissue showed that SAK can promote the activation of the innate immune inflammatory response including the NLRP3 inflammasome. In addition, SAK can also increase the content of cytokines in serum and BALF of mice after *S. aureus* infection. These features of inflammatory hyperactivation are consistent with stronger inflammatory cell infiltration and pulmonary edema in the lung tissue of wild-type ST398-infected mice.

The canonical NLRP3 inflammasome activation process is thought to require a priming step prior to activation. Various stimuli including bacterial toxins can activate the NLRP3 inflammasome, but no direct interaction between NLRP3 and these stimuli has been observed[44]. It seems that they may activate NLRP3 by inducing various cell signal transduction events, including K$^+$ efflux, ROS production and lysosomal damage[12]. Our results suggest that although SAK cannot directly activate the NLRP3 inflammasome, SAK may play a role in the priming step of NLRP3 inflammasome activation. This may be due to the fact that SAK can promote the activation of NF-κB, thereby indirectly up-regulating the expression levels of inflammasome-related molecules. Our further studies showed that SAK can promote ROS production and K$^+$ efflux, which are important upstream events of NLRP3 inflammasome activation, and may also be the mechanism by which SAK promotes NLRP3 inflammasome activation. Interestingly, although purified SAK played a modest role in priming the NLRP3 inflammasome, in the presence of an effective NLRP3 inflammasome activator (such as LPS), SAK may further promote Caspase-1 maturation and IL-1β release. This was also indicated by the ST398 secretion supernatant stimulated BMDMs to release higher levels of IL-1β compared to the *sak*

knockout strain. Due to the presence of many virulence factors in bacterial supernatants, SAK may have additive effects with other NLRP3 activators, ultimately promoting further activation of the NLRP3 inflammasome. In addition, SAK can also promote cytokine release and ROS production in human THP-1 cells, suggesting that the activation of host inflammatory response by SAK is also applicable to human infection. The mechanism by which SAK helps *S. aureus* evade host innate immunity by inhibiting antimicrobial peptides is different from SAK's promotion of inflammasome activation in this study, but both contribute to the promotion of *S. aureus* infection. Notably, the supernatant of the *sak* knockout strain could still activate the NLRP3 inflammasome, leading to pyroptosis and IL-1β release, which may be driven by other important virulence factors secreted by *S. aureus*. In fact, multiple studies have shown that α-toxin[45], Panton-Valentine leukocidin[46] and other *S. aureus* exocrine toxins can effectively promote the activation of the NLRP3 inflammasome. Although classical NLRP3 inflammasome activation is thought to be achieved through a two-step process of priming and activation, studies have shown that LPS alone can also directly activate NLRP3 through an unknown mechanism[47]. Furthermore, the cellular events of NLRP3 activation are complicated by the fact that NLRP3 activators are able to trigger multiple cellular signals and that there are also interactions between these signals[33].

The inhibition of NLRP3 may diminish pathology in *S. aureus* pulmonary infection. This may be due to the fact that lung infection is prone to severe pathological changes, and tissue destruction caused by excessive inflammation can lead to extremely impaired lung function[48]. Therefore, a balance of antimicrobial immunity and overall inflammation is required to reduce inflammatory damage to lung tissue and improve patient outcomes[45]. We emphasize that in the pneumonia caused by CA-SA, the control of inflammation is as important as the anti-infection strategy, and that inhibition of the NLRP3 inflammasome may serve as a mechanism for severe acute *S. aureus* pneumonia adjuvant therapy.

In this study, we investigate the pathogenic mechanism by which the pro-inflammatory effect of SAK promotes CA-SA infection in a mouse model of severe pneumonia. Previous studies pay more attention to the promoting effect of SAK on fibrinolysis.

Our study shows that SAK is also an effective pro-inflammatory factor for acute infections caused by CA-SA. Our results suggest that SAK may contribute to the activation of NF-κB and promote $K^+$ efflux and ROS production, thereby playing a role in the activation of the NLRP3 inflammasome. This enriches our understanding of the function of SAK, and also provides new ideas for the pathogenic mechanism of high virulence CA-SA.

## Methods

**Bacterial isolates**. *S. aureus* strains were isolated from adult patients of Renji Hospital affiliated with Shanghai Jiaotong University. The concentration of antibiotics used when constructing gene knockout strains is as follows: ampicillin, 100 μg/ml; chloramphenicol, 10 μg/ml. Please see the Supplementary methods for bacterial growth conditions and molecular typing. In addition, bacterial strains and plasmids used in this study can be found in Supplementary Table 1.

**Measurement of SAK secretion by *S. aureus***. Overnight cultures (12–15 h) were diluted 1:100 into fresh TSB and incubated at 37 °C with shaking at 200 rpm. Bacterial suspension at different time points was centrifuged (4000 rpm for 10 min at 4 °C) and the supernatant was collected after filtration. Human plasminogen (Roche) was mixed with supernatant and incubated at 37 °C for 30 min, then a plasmin-specific chromogenic substrate S-2251 (3 mmol/L, Chromogenix) was added and incubation was continued at 37 °C for 60 min. The amount of SAK secreted by the strains was read with a micro enzyme-linked immunosorbent assay (ELISA) autoreader (BioRad) at 405 nm.

**Semiquantitative biofilm assay**. Overnight cultures (12–15 h) were diluted into fresh TSB (with 0.5% glucose) to a final optical density of 0.1 (600 nm). The diluted *S. aureus* cultures were dispensed in 96-well flat-bottom tissue culture plates (200 μL/well), and incubated at 37 °C for 24 h without shaking. In some experiments, 10% plasma or purified SAK (2 μg/ml) were added into TSB medium (with 0.5% glucose). Heparin anticoagulated human plasma was provided voluntarily by healthy subjects. Mouse plasma came from the experimental C57BL/6 mice. Both human and mouse plasma were incubated at 56 °C for 30 min before use to inactivate complement. After incubation, the culture supernatant was gently removed and the wells were washed twice with sterile phosphate buffered saline (PBS). Bouin's fixative was added to the bottom of the well, and after 1 hour of fixation, the well was washed three times with PBS, and then the biofilm was stained with crystal violet. The floating stain was washed off with running water and the ability of biofilm formation was reflected by the absorbance at 570 nm (Micro ELISA autoreader, BioRad).

**Lung infection model in mice**. C57BL/6 wild-type (Shanghai JSJ Laboratory Animal Co., Ltd.) or *CRAMP*$^{−/−}$ (Jackson Laboratory) female mice (7–8 weeks old) were used for the lung infection model. *S. aureus* strains were grown to mid-logarithmic phase, washed once and then resuspended with sterile PBS at $5 × 10^6$ CFU/μl. Mice were anesthetized with 2,2,2-tribromoethanol (3.75–5 mg /25 g), and then 40 μl inoculum was pipetted into the nares of the anesthetized mice slowly. The mice were euthanized 48 h after infection, and their lungs were dissected out. The left lung was homogenized in 0.5 ml of TSB, the homogenized tissue was diluted and plated on TSB agar for CFU determination. The right lung was fixed in 4% formalin and tissue sections were prepared for HE staining and immunohistochemical staining. The remaining right lungs were used for RNA sequencing. For the fatal pneumonia model, 50 mg/kg MCC950 sodium (NLRP3 inflammasome inhibitor, Selleck) or sterile PBS was injected intraperitoneally into mice 2 h before infection by $10^9$ CFU ($2 × 10^7$ CFU/μl, 50 μl) *S. aureus*. Lung tissues were taken out immediately after the mouse died.

**Cell culture**. BMDMs were isolated from bone marrow of C57BL/6 wild-type or *Caspase-1*$^{−/−}$ (Jackson Laboratory) female mice (7–8 weeks old) and differentiated for 7 days in RPMI 1640 medium supplemented with 10% fetal bovine serum (Gibco), penicillin (100 U/ml) and streptomycin (0.1 mg/ml) and 20 ng/ml human M-CSF (R&D Systems). THP-1 (Cell Bank of Shanghai Institutes of Biological Sciences, Chinese Academy of Sciences) were cultured in RPMI 1640 medium with FBS and penicillin and streptomycin. Phorbol 12-myristate 13-acetate (100 ng/ml, 24 h) was used in differentiation of THP-1 into macrophages. The additives or inhibitors used in in vitro experiments are as follows: purified SAK (0.5 μg/ml), lipopolysaccharide (LPS, 0.2 μg/ml), ATP (5 mM), MCC950 (1 μM, Selleck), potassium channel blocker glibenclamide[49] (10 μM, Sigma), KCl (25 mM), NADPH oxidase inhibitor apocynin[50] (200 μM, Selleck), lysosome membrane stabilizer dexamethasone[51] (200 nM, Sangon Biotech), ROS scavenger NAC[52] (1 mM, Sigma), and NF-κB inhibitor BAY 11-7082 (10 μM, Sigma).

**Quantitative reverse-transcription (qRT) PCR**. Total RNA of *S. aureus* or cells was extracted, and complementary DNA was synthesized from total RNA by using the QuantiTect reverse transcription system (Qiagen). Amplification of the resulting complementary DNA sample utilized the QuantiTect SYBR green PCR kit

(Qiagen). The 7500 Sequence Detector (Applied Biosystems) was used to perform reactions in MicroAmp Optical 96-well reaction plates. Oligonucleotides used in this study are presented in Supplementary Table 2.

**RNA Sequencing**. Mice were euthanized 48 h after infection and their lungs were dissected. Total RNA was extracted from lung tissues (Qiagen). Nanodrop2000 was used to detect the concentration and purity of the extracted RNA, agarose gel electrophoresis was used to detect RNA integrity, and the Agilent 2100 was used to determine the RIN value. The library was constructed using the Illumina Truseq™ RNA sample prep Kit method, and finally sequenced on the Illumina Novaseq 6000 platform (Shanghai Majorbio Bio-pharm Technology Co., Ltd).

**Western blot**. Overnight cultures (12–15 h) were diluted 1:100 into fresh TSB and incubated at 37 °C with shaking until stationary phase (OD600$_{nm}$ = 5.0). The purified SAK (0.5 μg/ml) or LPS (0.2 μg/ml) or *S. aureus* culture filtrates (1/20-fold dilution) were added into mouse BMDMs cultured in a 6-well plate and incubate for 3 h. Some groups need to add 2.5 mM ATP to incubate for another 30 min. In some experiments, BMDMs were primed with LPS for 3 h, and then LPS-primed cells were stimulated with SAK (0.5 μg/ml, 30 min) or ATP (2.5 mM, 30 min, as a positive control) to determine whether purified SAK could activate the inflammasome. Cells were collected after being washed by sterile PBS, and boiled in loading buffer. Then the samples were subjected to western blot experiment with Caspase-1 (Abcam, ab207802, 1:1000), GSDMD (Abcam, ab209845, 1:1000) interleukin-1β (IL-1β, Abcam, ab234437, 1:1000), β-actin (Abcam, ab8227, 1:3000), NF-κB p65(Cell Signaling Technology, 8242, 1:1000), Phospho-NF-κB p65 (Cell Signaling Technology, 3033, 1:1000) antibodies. The densitometry of each band was quantified by Image J.

**Cytokine detection**. Anesthetize the mice with 2,2,2-tribromoethanol (3.75–5 mg/ 25 g), then slowly pipet the inoculum with $2 × 10^8$ CFU into the nostrils of the mice. The orbital venous plexus blood of mice was collected 48 h after infection, and the BALF was collected after euthanizing the mice. The level of mouse plasma cytokine was detected by Bio-Plex Pro Mouse Cytokine 23-plex Assay in accordance with the manufacturer's instructions (Bio-Rad, Luminex 200™ System). Cytokine levels in mouse BALF were detected by BD Cytometric Bead Array (CBA) Mouse Th1/Th2/Th17 Cytokine Kit. BD FACSCantoII was used for data collection and FCAP Array V3.0 was used for data analysis. IL-1β in the BALF and mouse BMDMs supernatants was measured by using ELISA kit (R&D systems). IL-1β and tumor necrosis factor α (TNF-α) released by THP-1 cells were detected by Human IL-1β or TNF-α ELISA Kit (Sangon Biotech), and interleukin-17A (IL-17A) released by THP-1 was measured by Human IL-17 Quantikine HS ELISA Kit (R&D systems).

**Cytotoxicity detection (LDH release)**. BMDMs were incubated for 3 h with purified SAK or a 1/20 or 1/100 dilution of each *S. aureus* supernatant (stationary phase). BMDMs lysis was measured using a lactate dehydrogenase (LDH) cytotoxicity assay kit according to the manufacturer's protocol (Roche). The released LDH was normalized to the total LDH content measured in BMDMs samples infiltrated with 2% Triton X-100.

**Measurement of intracellular $K^+$**. Cells were quickly washed and resuspended (by scraping the cells) in nuclease-free water for three freeze-thaw cycles. Lysates were centrifuged at 16,000 g for 10 min at 4 °C[53]. Supernatants were taken and $K^+$ concentrations were quantified by indirect potentiometric methods using a Cobas 8000 with ISE module (Automatic biochemistry analyzer, Roche).

**Detection of ROS**. SAK (0.5 μg/ml) was added into the medium of BMDMs or THP-1 cells, and NAC (1 mM, Sigma) or PBS was added at the same time in 12-well plates of cultured cells at 37 °C, and incubated for 5 h, after which the fluorescent dye 2,7-Dichlorodi-hydrofluorescein diacetate (DCFH-DA, Sigma) was added to the cell culture medium and incubated for 30 min at 37 °C in the dark. After washing off the excess dye, the fluorescence signal was collected at an excitation wavelength of 480 nm and emission wavelength of 525 nm, and the cells placed under a fluorescence microscope for observation.

**Statistics and reproducibility**. Statistical analyses were performed with Graph-Pad Prism, version 8.0. The comparison of survival curves was carried out by log-rank (Mantel-Cox) test. Unpaired $t$ tests and two-way ANOVA were used to compare the differences between the variables. All error bars in the graphs show standard deviation (±SD). $P < 0.05$ was regarded as statistically significant. Sample sizes and replicates are described in the corresponding legends.

**Ethics statement**. All animal experiments were performed in accordance with the laboratory animal care and use guidelines of the Chinese Association for Laboratory Animal Sciences (CALAS). Heparinized venous blood was donated voluntarily by healthy subjects and *S. aureus* strains were isolated from patients. All participants or their legal guardians have provided written informed consent to take part

in the study. This study was approved by the ethics committee of Renji Hospital, School of Medicine, Shanghai Jiao Tong University, Shanghai (RA-2020-229).

## Data availability

RNA-sequence data of this study have been submitted to NCBI (Accession: PRJNA767166, Sample: SRR16107127, SRR16107128, SRR16107129, SRR16107130). Details of methods are described in Supplementary Methods. Top 50 significantly different genes in RNA-seq analysis of mouse lung tissue are showed in Supplementary data 1. The source data underlying the figures presented in this manuscript are provided in Supplementary data 2. Extra data are available from the corresponding author upon request.

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

## Acknowledgements

This work was supported by the National Natural Science Foundation of China (82102455, 81861138043, 81800657), the Shanghai Committee of Science and Technology, China (19JC1413005), Clinical Research Plan of SHDC (SHDC2020CR3006A) and Shanghai Sailing Program (21YF1425500).

## Author contributions

Y.W. and N.Z. performed the experiments and wrote the manuscript. Y.J. and Y.L. analyzed the RNA-sequence data. L.Z. and L.H. performed the statistical analysis. Q.L. and M.L. designed the experiments and revised the manuscript. All authors reviewed the manuscript.

## Competing interests

The authors declare no competing interests.
