## [Peer Review file · Communications Biology]

Reviewers' comments:

Reviewer #1 (Remarks to the Author):

This is an interesting study by Wang et al. The authors highlight the high prevalence of the gene sak, which encodes the protein staphylokinase (Sak), within the genomes of human-adapted *S. aureus* strains (both hospital- and community-associated). Despite the collective high genomic carriage of sak by hospital (HA-SA) and community-associated (CA-SA) strains, the latter appear to have increased sak expression, concurrent with low biofilm forming capacity, which the authors attribute to high Agr activity.

Importantly, the authors investigate how Sak from CA-SA promotes pneumonia via a mechanism independent of its well characterized fibrinolytic or antimicrobial inhibiting effects. They show that Sak induces reactive oxygen species (ROS) production which in turn stimulates the NLRP3 inflammasome, triggering acute infection.

The data are new and the study well executed and timely given the prior heavy research focus on the plasminogen-activating properties of Sak that promote fibrinolysis and invasion into deeper tissues. I only have minor comments.

Minor comments:

1. The abstract could be tweaked to highlight the main message more clearly/succinctly so as to immediately captivate readers. For example, the part pertaining to the observation of the same phenotype in WT and CRAMP^{-/-} mice could be reworded to emphasize that the novel virulence mechanism of Sak is independent of its ability to inhibit CRAMP in this mouse model.
2. ST398 is one of the most prevalent CA-SA clones in China but how representative of the hospital-associated strains are ST5 and ST239?
3. Fig. 1A - Whilst the isolates have been labelled as human-adapted (HO) versus livestock-adapted (LA), it would be good to also note underneath the HO strains, which ones are hospital- versus community-acquired as in Figure 1B (HA and CA) for easier reading/interpretation of the manuscript.
4. The authors show that community-acquired *S. aureus* (CA-SA) have high sak genomic prevalence, gene expression and production of the protein (Fig. 1A-C), but low biofilm forming ability (Fig. 1D). However, they do not show the corresponding growth curve controls. Do these strains grow similarly? If not, the biofilm should be displayed as OD_{570nm}/OD_{600nm} instead.
5. Lines 136-137 - add "as shown by increased staining of F4/80, a well characterized macrophage marker".
6. Could the top 50 significantly different genes at least be displayed in the supplementary results? Were inflammatory cytokines also upregulated in the mouse BALF during infection with the WT strain as compared to the Δ sak mutant?
7. Lines 159-160, 163-165 - please be mindful that the "more active state of inflammatory response" of WT ST398-infected mice is consistent with the higher bacterial burden.
8. Lines 166-183 - Is 0.5 μ g within the physiological range of secreted Sak during infection? Are the authors saying that the Sak protein (by itself) is mainly providing signal 1 for inflammasome activation (signal 1- transcriptional upregulation of pro-IL1 β , pro-IL18 and molecules of the inflammasome machinery) via ROS and the induction of transcription factors such as NF κ B? Could the western blot (particularly Fig. 4G) please be accompanied by densitometry?

Reviewer #2 (Remarks to the Author):

The manuscript by Wang and Zhao et al. reports the role of Staphylokinase (SAK) in promotion of lung infection in WT and mice deficient in antimicrobial peptide CRAMP. Further the authors found that SAK can promote activation of the NLRP3 inflammasome.

Major comments:

1. Fig4A: The reduction in expression of NLRP3 and cytokines such as IL-1Beta, IL-17, TNF and IL-8 in ST398 SAK KO vs WT indicate that SAK KO bacteria might lack certain priming signals. Due to expression difference of NLRP3 in WT vs SAK KO ST398 it is questionable whether the secretion of IL-1B (Fig 4H) is due SAK induced inflammasome activation or the inability of the SAK KO ST398 bacteria to induce expression of NLRP3 and cytokines. The authors should address this concern since this data is key to prove that SAK activates the NLRP3 inflammasome.
2. Fig 4G: The authors should provide an explanation of using LPS+ATP +SAK as a treatment group. How will the authors determine the contribution of inflammasome activation by ATP and SAK? Please revise this experiment. This treatment group makes no sense and should be removed. Instead, the authors should consider priming the BMDMs for 3h with LPS and then stimulate the LPS-primed cells with SAK for determining whether purified SAK can activate the inflammasome.
3. To prove whether SAK can activate the NLRP3 inflammasome, the authors should use NLRP3 KO mice and BMDMs harvested from these mice for providing genetic evidence of NLRP3 inflammasome activation.
4. It is interesting to note that SAK KO ST398 bacteria can still induce caspase-1 cleavage and secretion of IL1Beta indicating that other virulence factor can induce activation of the inflammasome. The authors should discuss this finding and put forward the literature on other virulence factors of *S. aureus* capable of activating the inflammasome.
5. Fig 4G- Please provide the blot of the cleaved IL-1Beta.
6. Use high extracellular concentration of K ions (PMID: 23809161) in addition of using glibenclamide.
7. Line 215- The figure 6D indicated NS in expression of IL-1Beta in the presence of apocynin, dexamethasone and NAC. Please revise this sentence and inference.

Minor comments:

1. The explanation of Fig 6D-E in the text between lines 215-219 is very confusing and will recommend simplifying the inference for ease of reading.
2. Line 193- for ease of readership please infer as- 'survival was prolonged'.

Reviewer #3 (Remarks to the Author):

Excessive inflammatory responses are associated with pneumonia in pulmonary infections like influenza, SARS-CoV-2, and *S. aureus*. Studying mechanisms underlying these excessive inflammatory responses is a current research topic that will shed light on mitigating pulmonary pathologies. This study investigates the role of staphylokinase (SAK) in community-associated *S. aureus*-induced pneumonia. The authors established SAK's role in promoting excessive inflammation and lung infection after *S. aureus* infection (ST398 strain). In vivo experiments demonstrate fibrinolysis independent functions of SAK in regulating NLRP3 inflammasome activation and inflammatory cell infiltration in the lung. With the limited experimental data, the authors claim that SAK-mediated ROS signaling promotes NLRP3 inflammasome and causes excessive inflammation. Although the study is interesting and establishes the SAK role in *S. aureus* induced pathogenic inflammation, the proposed NLRP3 inflammasome mechanism is poorly demonstrated and does not explore other NLRP3 inflammasome independent functions. Specific comments are mentioned below.

1. Experiments in this study do not support the direct role of SAK in NLRP3 inflammasome activation. RNA sequencing data clearly indicates SAK role in the upregulation of inflammatory genes. RNA sequencing, immunoblotting and IL-1b ELISA studies suggest that SAK might have a role in NLRP3 inflammasome priming (signal 1) instead of activation (signal 2). The CASP1 activation (p20 levels) correlates with NLRP3 and CASP1 expression differences seen in the presence and absence of SAK.

Also, CASP1 activation differences are observed only when BMDMs/THP-1 cells were challenged with the ST398 strain, and SAK expression alone did not induce CASP1 activation. Based on the data provided in this manuscript, SAK likely promotes NF- κ B activation and upregulation of inflammasome and other pro-inflammatory genes (but may not inflammasome activation). SAK role on NLRP3 inflammasome is an indirect effect. Authors need to probe for NF- κ B and other innate immune transcription factors activation to identify a specific role for SAK. Does the absence of SAK show less cell death (in particular, pyroptosis) in *S. aureus*-infected cells?

2. Several studies indicate SAK's role in evading innate immune responses. Authors need to justify how this work significantly differs from existing studies.

3. In Figure-2, it was mentioned that severe inflammation and increased bacterial load is seen 48 hr after infection. In Figure-3, what time the lung samples were collected for RNA sequencing after infection? Are these samples collected 48 hr post-infection or collected at earlier time points? These details are missing in the figure legend and methods section. This information is critical to correlate the role of SAK in priming inflammation or propagating inflammatory signaling.

4. Essential details of the experimental setup are missing in Figure-5 and Figure-6. What time were cell lysates collected for immunoblotting analysis and ELISA studies?

Response to reviewers:

Many thanks to the reviewers for their interests in our work and their insightful and constructive comments and criticisms. Accordingly, we have performed a large number of new experiments and extensively revised the manuscript. We have also expanded the Discussion sections. Please find below our responses to all comments and criticisms. The data generated during the revision are highlighted in red color in the manuscript.

Reviewer #1 (Remarks to the Author):

This is an interesting study by Wang et al. The authors highlight the high prevalence of the gene *sak*, which encodes the protein staphylokinase (Sak), within the genomes of human-adapted *S. aureus* strains (both hospital- and community-associated). Despite the collective high genomic carriage of *sak* by hospital (HA-SA) and community-associated (CA-SA) strains, the latter appear to have increased *sak* expression, concurrent with low biofilm forming capacity, which the authors attribute to high Agr activity.

Importantly, the authors investigate how Sak from CA-SA promotes pneumonia via a mechanism independent of its well characterized fibrinolytic or antimicrobial inhibiting effects. They show that Sak induces reactive oxygen species (ROS) production which in turn stimulates the NLRP3 inflammasome, triggering acute infection.

The data are new and the study well executed and timely given the prior heavy research focus on the plasminogen-activating properties of Sak that promote fibrinolysis and invasion into deeper tissues. I only have minor comments.

Response: We greatly appreciate the reviewer for his/her positive and enthusiastic comments.

Minor comments:

1. The abstract could be tweaked to highlight the main message more clearly/succinctly so as to immediately captivate readers. For example, the part pertaining to the observation of the same phenotype in WT and CRAMP^{-/-} mice could be reworded to emphasize that the novel virulence mechanism of Sak is independent of its ability to inhibit CRAMP in this mouse model.

Response: Thanks for the reviewer's insightful suggestion. We have revised the Abstract section to emphasize that the effects of SAK are not entirely dependent on CRAMP, and to highlight the role of inflammasome activation in SAK's promotion of *S. aureus* pathogenesis.

2. ST398 is one of the most prevalent CA-SA clones in China but how representative of the hospital-associated strains are ST5 and ST239?

Response: Thanks to the reviewers for the comment. According to the results of

epidemiological studies, ST5 and ST239 are the most prevalent clones of hospital-associated *S. aureus* in China. We have included this information in the introduction section of the revised manuscript.

3. Fig. 1A - Whilst the isolates have been labelled as human-adapted (HO) versus livestock-adapted (LA), it would be good to also note underneath the HO strains, which ones are hospital- versus community-acquired as in Figure 1B (HA and CA) for easier reading/interpretation of the manuscript.

Response: Thanks to the reviewer's suggestion, we have added these details and labelled the source of the human-adapted isolates.

4. The authors show that community-acquired *S. aureus* (CA-SA) have high sak genomic prevalence, gene expression and production of the protein (Fig. 1A-C), but low biofilm forming ability (Fig. 1D). However, they do not show the corresponding growth curve controls. Do these strains grow similarly? If not, the biofilm should be displayed as OD570nm/OD600nm instead.

Response: Thanks to the reviewers for the comment. As suggested, we have supplemented the growth curves of these clinical strains, and the results showed that the growth trends of these strains were basically the same (Supplementary Figure 1a). In addition, semiquantitative biofilm experiments in this study were performed by incubating diluted cultures (with the same optical density) at 37°C for 24 hours without shaking.

5. Lines 136-137 – add “as shown by increased staining of F4/80, a well characterized macrophage marker”.

Response: By following the reviewer’s comment, we have added the description of F4/80.

6. Could the top 50 significantly different genes at least be displayed in the supplementary results? Were inflammatory cytokines also upregulated in the mouse BALF during infection with the WT strain as compared to the Δsak mutant?

Response: According to the reviewer’s suggestion, we have presented the top 50 significantly different genes in Supplementary Data 1. In addition, we have also detected 8 cytokines in mouse BALF, and the results showed that most cytokines (IL-6, IL-10, TNF- α , IFN- γ and IL-1 β) were at higher levels in mouse BALF infected with wild-type ST398 (Figure 3c, d).

7. Lines 159-160, 163-165 – please be mindful that the “more active state of inflammatory response” of WT ST398-infected mice is consistent with the higher bacterial burden.

Response: Thanks to the reviewer's suggestion, we have added this in the revised manuscript.

8. Lines 166-183 - Is 0.5 μg within the physiological range of secreted Sak during

infection?

Are the authors saying that the Sak protein (by itself) is mainly providing signal 1 for inflammasome activation (signal 1- transcriptional upregulation of pro-IL1b, pro-IL18 and molecules of the inflammasome machinery) via ROS and the induction of transcription factors such as NFκB? Could the western blot (particularly Fig. 4G) please be accompanied by densitometry?

Response: We appreciate these insightful comments from the reviewer. We have quantified SAK levels in stationary phase (12 h) supernatants by using standard dilutions of recombinant SAK and found that the concentration of SAK secreted by CA-SA was approximately 3 μg/ml (Supplementary Figure 1b), therefore, we considered 0.5μg/ml dose not exceeding the physiological range of SAK.

To explore whether SAK plays a role in the priming or activation step of the NLRP3 inflammasome activation, we have performed a large number of new experiments and extensively revised the manuscript. Our results suggest that although SAK cannot directly activate the NLRP3 inflammasome, SAK may play a role in the priming step of NLRP3 inflammasome activation (Figure 4, Supplementary Figure 3). This may be due to the fact that SAK can promote the activation of NF-κB (Figure 6, Supplementary Figure 5), thereby indirectly up-regulating the expression levels of inflammasome-related molecules. Our further studies showed that SAK can promote ROS production and K⁺ efflux (Figure 6), which are important upstream events of NLRP3 inflammasome activation, and may also be the mechanism by which SAK promotes NLRP3 inflammasome activation. In addition, the protein expression levels in figure 4 were quantified by relative densitometric analysis and shown in Supplementary Figure 3.

Reviewer #2 (Remarks to the Author):

The manuscript by Wang and Zhao et al. reports the role of Staphylokinase (SAK) in promotion of lung infection in WT and mice deficient in antimicrobial peptide CRAMP. Further the authors found that SAK can promote activation of the NLRP3 inflammasome.

Major comments:

1. Fig4A: The reduction in expression of NLRP3 and cytokines such as IL-1Beta, IL-17, TNF and IL-8 in ST398 SAK KO vs WT indicate that SAK KO bacteria might lack certain priming signals. Due to expression difference of NLRP3 in WT vs SAK KO ST398 it is questionable whether the secretion of IL-1B (Fig 4H) is due SAK induced inflammasome activation or the inability of the SAK KO ST398 bacteria to induce expression of NLRP3 and cytokines. The authors should address this concern since this data is key to prove that SAK activates the NLRP3 inflammasome.

Response: We appreciate the reviewer's insightful comment. As suggested, we have performed a large number of new experiments and extensively revised the manuscript.

We found that SAK may play a role in the priming step of NLRP3 inflammasome activation (Figure 4, Supplementary Figure 3). Although the knockout of *sak* reduced the expression of inflammation-related genes in ST398 supernatant-stimulated macrophages, the expression levels of these genes were still significantly increased compared with negative control or purified SAK-stimulated macrophages (Figure 4 a-e). This may be driven by other virulence factors secreted by *S. aureus*. However, when using NLRP3 inhibitor or knocking out the *caspase-1* gene in mice, ST398-WT and *sak* knockout strains no longer had significant differences in the release of IL-1 β from macrophages (Figure 4l, Supplementary Figure 3i). Taken together, our results indicate that the effect of SAK on the release of IL-1 β from macrophages may be more mediated through NLRP3 inflammasome activation.

2. Fig 4G: The authors should provide an explanation of using LPS+ATP +SAK as a treatment group. How will the authors determine the contribution of inflammasome activation by ATP and SAK? Please revise this experiment. This treatment group makes no sense and should be removed. Instead, the authors should consider priming the BMDMs for 3h with LPS and then stimulate the LPS-primed cells with SAK for determining whether purified SAK can activate the inflammasome.

Response: We appreciate these insightful comments from the reviewer. As suggested, we modified the experimental design based on your suggestion and added experiments to explore the role of SAK in the priming and activation steps of NLRP3 inflammasome activation (Figure 4, Supplementary Figure 3).

3. To prove whether SAK can activate the NLRP3 inflammasome, the authors should use NLRP3 KO mice and BMDMs harvested from these mice for providing genetic evidence of NLRP3 inflammasome activation.

Response: The reviewer's suggestion is greatly appreciated, unfortunately we were unable to obtain NLRP3 knockout mice. But we achieved a similar effect by applying an NLRP3 inhibitor to the BMDMs from wild-type mouse (Supplementary Figure 3g-i), and we also performed experiments using the BMDMs from *caspase-1*^{-/-} mouse (Figure 4k, l).

4. It is interesting to note that SAK KO ST398 bacteria can still induce caspase-1 cleavage and secretion of IL1Beta indicating that other virulence factor can induce activation of the inflammasome. The authors should discuss this finding and put forward the literature on other virulence factors of *S. aureus* capable of activating the inflammasome.

Response: We appreciate the reviewer mentioning this interesting data. Indeed, multiple studies have shown that α -toxin, Panton-Valentine leukocidin and other *S. aureus* exocrine toxins can effectively promote the activation of the NLRP3 inflammasome. We have discussed this finding and annotated the literature that other *S. aureus* virulence factors are able to activate the inflammasome.

5. Fig 4G- Please provide the blot of the cleaved IL-1Beta.

Response: We have now added the blot of the cleaved IL-1 β in the revised Fig 4.

6. Use high extracellular concentration of K ions (PMID: 23809161) in addition of using glibenclamide.

Response: As suggested, we have used high extracellular concentration of K ions to treated BMDMs and found that SAK no longer exhibited additional effects on promoting IL-1 β expression and release (Fig 6a, b). In addition, we have also examined intracellular K⁺ levels and found that the presence of SAK promoted K⁺ efflux (Fig 6c).

7. Line 215- The figure 6D indicated NS in expression of IL-1Beta in the presence of apocynin, dexa and NAC. Please revise this sentence and inference.

Response: Thanks to the reviewer's suggestion, we have revised this section of the results description and inferences.

Minor comments:

1. The explanation of Fig 6D-E in the text between lines 215-219 is very confusing and will recommend simplifying the inference for ease of reading.

Response: We apologize for this confusing explanation, we have supplemented and re-described this part of the results.

2. Line 193- for ease of readership please infer as- ‘survival was prolonged’.

Response: Thanks to the reviewers for pointing this out. We have rephrased the sentence, and inferred it as “survival was prolonged”.

Reviewer #3 (Remarks to the Author):

Excessive inflammatory responses are associated with pneumonia in pulmonary infections like influenza, SARS-CoV-2, and *S. aureus*. Studying mechanisms underlying these excessive inflammatory responses is a current research topic that will shed light on mitigating pulmonary pathologies. This study investigates the role of staphylokinase (SAK) in community-associated *S. aureus*-induced pneumonia. The authors established SAK’s role in promoting excessive inflammation and lung infection after *S. aureus* infection (ST398 strain). In vivo experiments demonstrate fibrinolysis independent functions of SAK in regulating NLRP3 inflammasome activation and inflammatory cell infiltration in the lung. With the limited experimental data, the authors claim that SAK-mediated ROS signaling promotes NLRP3 inflammasome and causes excessive inflammation. Although the study is interesting and establishes the SAK role in *S.aureus* induced pathogenic inflammation, the proposed NLRP3 inflammasome

mechanism is poorly demonstrated and does not explore other NLRP3 inflammasome independent functions. Specific comments are mentioned below.

1. Experiments in this study do not support the direct role of SAK in NLRP3 inflammasome activation. RNA sequencing data clearly indicates SAK role in the

upregulation of inflammatory genes. RNA sequencing, immunoblotting and IL-1 β ELISA studies suggest that SAK might have a role in NLRP3 inflammasome priming (signal 1) instead of activation (signal 2). The CASP1 activation (p20 levels) correlates with NLRP3 and CASP1 expression differences seen in the presence and absence of SAK. Also, CASP1 activation differences are observed only when BMDMs/THP-1 cells were challenged with the ST398 strain, and SAK expression alone did not induce CASP1 activation. Based on the data provided in this manuscript, SAK likely promotes NF- κ B activation and upregulation of inflammasome and other pro-inflammatory genes (but may not inflammasome activation). SAK role on NLRP3 inflammasome is an indirect effect. Authors need to probe for NF- κ B and other innate immune transcription factors activation to identify a specific role for SAK. Does the absence of SAK show less cell death (in particular, pyroptosis) in *S. aureus*-infected cells?

Response: We appreciate this insightful comment from the reviewer. We have performed a large number of new experiments and extensively revised the manuscript. We have assessed cell death by measuring LDH released by BMDMs after co-incubation with SAK or *S. aureus* supernatants, and the results showed that *sak* gene knockout significantly reduced macrophage death (Figure 4f). Moreover, we have also examined the role of SAK in NLRP3 priming and found that cleavage of caspase-1 and Gasdermin D (GSDMD, an important protein that mediates pyroptosis) and release of IL-1 β could be observed in SAK-treated cells after the addition of ATP, although the effect was not as strong as that of LPS (Figure 4i, j, Supplementary Figure 3b). In addition, we have primed the BMDMs for 3h with LPS and then treated the LPS-primed cells with SAK for determining whether purified SAK can activate NLRP3 inflammasome. Our results suggest a limited role for SAK in the activation step of NLRP3 inflammasome (Supplementary Figure 3c-e). Furthermore, our results suggest that SAK may contribute to the activation of NF- κ B (Figure 6e, Supplementary Figure 5a, b) and promote K⁺ efflux and ROS production (Figure 6a-d), thereby leading to the activation of the NLRP3 inflammasome.

2. Several studies indicate SAK's role in evading innate immune responses. Authors need to justify how this work significantly differs from existing studies.

Response: As suggested, we have compared the role of SAK in evading innate immune responses and promoting NLRP3 inflammasome activation in the Discussion section of the manuscript. Antimicrobial peptides act primarily in the early stages of infection and are thought to be essential innate immune effectors against bacterial infections. *S. aureus* has a variety of factors that promote infection, and different infection sites trigger different immune responses in the host. Research shows that NLRP3 signaling plays an important role in the body's defense against skin and soft tissue infections. However, in acute pneumonia caused by highly virulent CA-SA, inflammasome signaling is not necessary for clearance of *S. aureus* from mouse lung tissue, and even leads to inflammation-related cell death. The inhibition of NLRP3 may diminish pathology in *S. aureus* pulmonary infection. This may be due to the fact that lung infection is prone to severe pathological changes, and tissue destruction caused by excessive inflammation can lead to extremely impaired lung function. Therefore, a

balance of antimicrobial immunity and overall inflammation is required to reduce inflammatory damage to lung tissue and improve patient outcomes. The mechanism by which SAK helps *S. aureus* evade host innate immunity by inhibiting antimicrobial peptides is different from SAK's promotion of inflammasome activation in this study, but both contribute to the promotion of *S. aureus* infection.

3. In Figure-2, it was mentioned that severe inflammation and increased bacterial load is seen 48 hr after infection. In Figure-3, what time the lung samples were collected for RNA sequencing after infection? Are these samples collected 48 hr post-infection or collected at earlier time points? These details are missing in the figure legend and methods section. This information is critical to correlate the role of SAK in priming inflammation or propagating inflammatory signaling.

Response: Thanks to the reviewers for pointing out this missing information. Lung samples were collected 48 hours after infection of mice, and total RNA was extracted from lung tissue for RNA sequencing. We have supplemented these details in the Figure legend and Methods section.

4. Essential details of the experimental setup are missing in Figure-4 and Figure-6. What time were cell lysates collected for immunoblotting analysis and ELISA studies?

Response: We apologize for this unclear experimental description, which has now been corrected. In figure 4, BMDMs were incubated with bacterial culture supernatant (stationary phase, 1:20 dilution) or SAK (0.5µg/ml) for 3 hours, and cell lysates or culture supernatants were collected for western blot and ELISA studies. While, in figure 6, BMDMs were pretreated with these inhibitors for 30 minutes and then incubated with SAK (0.5µg/ml) or bacterial secretion supernatant for 3 hours (stationary phase, 1:20 dilution). These details of the experimental setup have been presented in the Legend and Methods section.

REVIEWERS' COMMENTS:

Reviewer #1 (Remarks to the Author):

All of my previous comments have been addressed by the authors in their response. As per my previous comments, the manuscript is interesting and the findings novel, particularly in light of prior research focusing mainly on the fibrinolytic activity of Sak.

This time round when I read the revised manuscript though, I noticed the following, all of which are rather minor comments and do not negatively affect my recommendation.

1. The overall message is that Sak promotes the priming step (Signal 1, i.e., upregulation of pro-IL-1B and NLRP3 proteins) of NLRP3 inflammasome activation.

A. If Sak induces signal 1 (Fig 4a-c), why is it that similar levels of pro-IL1B are observed in the lysate of BMDMs incubated with the bacterial supernatant of delta sak mutant (Fig 4g), same timepoint of 3h for these experiments ?

B. Lines 199-201 (Page 10) sound a bit confusing - "examined the role of SAK in NLRP3 priming and found that the cleavage of caspase-1..." Please note that NLRP3 priming (signal 1) comprises the activation of NFkB that leads to the upregulation of pro-IL-1B and NLRP3 proteins whereas cleavage of GSDMD and pro-IL1B by caspase 1 is actually signal 2.

2. The complemented sak strain was not used in the in vivo models. I wonder whether the authors tried using the complemented strain in mice but had issues with plasmid stability in vivo. Having said that, the authors did make sure to use the complemented strains in vitro.

3. Line 28 (page 2) "but the underlying mechanisms remain largely unknown" could be a bit misleading given the wealth of information on how *S. aureus* causes disease (virulence factors including toxins, etc). Perhaps replace with something along the lines of "mechanisms of pathogenesis are still being investigated".

Reviewer #2 (Remarks to the Author):

The authors have addressed my questions and concerns and I recommend manuscript for publication.

Reviewer #3 (Remarks to the Author):

This revised version from Wang et al., substantially improved the manuscript based on this reviewer's comments. The new data provided in revised manuscript justifies the role of SAK in NLRP3 inflammasome activation and also in priming NLRP3 inflammasome activation.

Response to reviewers:

The co-authors and I would like to thank you for the time and effort spent in reviewing the manuscript. We have carefully considered the suggestion of Reviewers and make some changes. Please find below our responses to all comments and criticisms. The data generated during the revision are highlighted in red color in the manuscript.

Responds to the reviewers' comments:

Reviewer #1 (Remarks to the Author):

All of my previous comments have been addressed by the authors in their response. As per my previous comments, the manuscript is interesting and the findings novel, particularly in light of prior research focusing mainly on the fibrinolytic activity of Sak.

This time round when I read the revised manuscript though, I noticed the following, all of which are rather minor comments and do not negatively affect my recommendation.

1. The overall message is that Sak promotes the priming step (Signal 1, i.e., upregulation of pro-IL-1B and NLRP3 proteins) of NLRP3 inflammasome activation.
A. If Sak induces signal 1 (Fig 4a-c), why is it that similar levels of pro-IL1B are observed in the lysate of BMDMs incubated with the bacterial supernatant of delta *sak* mutant (Fig 4g), same timepoint of 3h for these experiments?

Response: We are very grateful to reviewer for carefully reviewing our manuscript and mentioning these interesting data. We did observe that BMDM lysates incubated with bacterial supernatants of delta *sak* mutant and wild-type strains exhibited similar levels of pro-IL-1 β . We speculate that there may be multiple reasons for this result. First, gene expression is affected by time intervals and regulatory factors at the transcriptional and translational stages, and their expression trends are not always consistent. Furthermore, although purified SAKs play a modest role in priming the NLRP3 inflammasome, in the presence of potent NLRP3 inflammasome activators such as LPS, SAKs may further promote caspase-1 maturation and IL-1 β release (Fig. 4i). Due to the presence of many virulence factors in bacterial supernatants, SAK may have additive effects with other NLRP3 activators, ultimately promoting the activation of the NLRP3 inflammasome and the cleavage of pro-IL-1 β into mature IL-1 β . The maturation of IL-1 β is a dynamic process, and most of the mature IL-1 β has been released into the culture. The cleavage of pro-IL-1 β to mature IL-1 β may be faster than the protein synthesis of pro-IL-1 β , which may be one of the reasons why similar levels of pro-IL-1 β were observed in BMDM lysates. Therefore, the presence of SAK does promote more pro-IL-1 β production (a portion of which has been cleaved into IL-1 β). In addition, we have revised the description and discussion of the experimental results in this part of the manuscript for better reading and understanding.

- B. Lines 199-201 (Page 10) sound a bit confusing - "examined the role of SAK in NLRP3 priming and found that the cleavage of caspase-1..." Please note that NLRP3

priming (signal 1) comprises the activation of NF κ B that leads to the upregulation of pro-IL-1 β and NLRP3 proteins whereas cleavage of GSDMD and pro-IL1 β by caspase 1 is actually signal 2.

Response: We apologize for this confusing explanation, we have supplemented and re-described this part of the results.

2. The complemented sak strain was not used in the *in vivo* models. I wonder whether the authors tried using the complemented strain in mice but had issues with plasmid stability *in vivo*. Having said that, the authors did make sure to use the complemented strains *in vitro*.

Response: Thanks to the reviewer for pointing this out. Complementary strains have plasmid stability issues mentioned by the reviewer in mouse models, which reduce the reproducibility of results. Therefore, we made sure to use the complemented strains in our *in vitro* experiments.

3. Line 28 (page 2) “but the underlying mechanisms remain largely unknown” could be a bit misleading given the wealth of information on how *S. aureus* causes disease (virulence factors including toxins, etc). Perhaps replace with something along the lines of “mechanisms of pathogenesis are still being investigated”.

Response: Thanks to the reviewer for pointing out this misleading description, and we have re-described this part.

Reviewer #2 (Remarks to the Author):

The authors have addressed my questions and concerns and I recommend manuscript for publication.

Response: We greatly appreciate the reviewer for his/her positive and enthusiastic comments.

Reviewer #3 (Remarks to the Author):

This revised version from Wang et al., substantially improved the manuscript based on this reviewer's comments. The new data provided in revised manuscript justifies the role of SAK in NLRP3 inflammasome activation and also in priming NLRP3 inflammasome activation.

Response: We are grateful to this reviewer for taking the time to read our manuscript and give us very helpful comments.

Sincerely,

Min Li, PhD

Department of Laboratory Medicine, Ren ji Hospital, Shanghai Jiao Tong University School of Medicine.

Email: ruth_limin@126.com

Tel: 86-21-68383297